# Ferroptosis in Cancer Cell Biology

**DOI:** 10.3390/cancers12010164

**Published:** 2020-01-09

**Authors:** Christina M. Bebber, Fabienne Müller, Laura Prieto Clemente, Josephine Weber, Silvia von Karstedt

**Affiliations:** 1Department of Translational Genomics, Medical Faculty, University of Cologne, Weyertal 155b, 50931 Cologne, Germany; christina.bebber@uk-koeln.de (C.M.B.); fabienne.mueller@uk-koeln.de (F.M.); lprietoc@uni-koeln.de (L.P.C.); jweber38@smail.uni-koeln.de (J.W.); 2Cologne Excellence Cluster on Cellular Stress Response in Aging-Associated Diseases (CECAD), Medical Faculty, University of Cologne, Joseph-Stelzmann-Straße 26, 50931 Cologne, Germany; 3Department I of Internal Medicine, University Hospital of Cologne, Kerpener Straße 62, 50937 Cologne, Germany

**Keywords:** ferroptosis, cancer, cell death, GPX4, inflammation

## Abstract

A major hallmark of cancer is successful evasion of regulated forms of cell death. Ferroptosis is a recently discovered type of regulated necrosis which, unlike apoptosis or necroptosis, is independent of caspase activity and receptor-interacting protein 1 (RIPK1) kinase activity. Instead, ferroptotic cells die following iron-dependent lipid peroxidation, a process which is antagonised by glutathione peroxidase 4 (GPX4) and ferroptosis suppressor protein 1 (FSP1). Importantly, tumour cells escaping other forms of cell death have been suggested to maintain or acquire sensitivity to ferroptosis. Therefore, therapeutic exploitation of ferroptosis in cancer has received increasing attention. Here, we systematically review current literature on ferroptosis signalling, cross-signalling to cellular metabolism in cancer and a potential role for ferroptosis in tumour suppression and tumour immunology. By summarising current findings on cell biology relevant to ferroptosis in cancer, we aim to point out new conceptual avenues for utilising ferroptosis in systemic treatment approaches for cancer.

## 1. Introduction

The emergence of electron transport chains as a means to generate a chemical gradient for the generation of adenosine triphosphate (ATP) is as ancient as the first single-celled organism undergoing photosynthesis to produce oxygen. Billions of years of evolution later, multicellular aerobic organisms generate ATP mainly by oxidative phosphorylation (OXPHOS) at the expense of atmospheric oxygen [1]. Herein, protein complexes belonging to the electron transport chain are located in the inner mitochondrial membrane where they transport electrons derived from nicotinamide adenine dinucleotide hydrogen (NADH) via redox reduction to the terminal electron acceptor oxygen (O_2_), which is thereby reduced to water (H_2_O). NADH is one of the major products of the tricarboxylic acid (TCA) cycle within the mitochondrial lumen driven by metabolites such as Acetyl-CoA derived from glycolysis and catabolic fatty acid oxidation (β-oxidation) [1]. During electron transport, a small proportion of electrons leak out and react with oxygen molecules to generate highly reactive superoxide (O∙^−2^) [2]. Superoxides can be transported into the cytosol using the mitochondrial permeability transition pore (mPTP) in the outer mitochondrial membrane. Thereby, OXPHOS is a major source of reactive oxygen species (ROS) in aerobic cells which can cause aberrant oxidation of proteins, lipids and DNA.

Therefore, cells have evolved to develop a complex cellular antioxidant defence system to ensure cellular survival. As such, superoxide dismutases (SOD1), localised either in the cytoplasm or in the mitochondrial matrix (SOD2), catalyse the dismutation of superoxides (O∙^−2^) generated during OXPHOS into slightly less reactive hydrogen peroxide (H_2_O_2_) and water (H_2_O) [2]. H_2_O_2_ is then further reduced by catalases, glutathione peroxidases (GPXs) or peroxiredoxins (PRDXs) [3]. Many antioxidant defence proteins including SOD1, catalase and glutathione peroxidase 4 (GPX4) [4] are transcriptionally induced by the antioxidant transcription factor nuclear factor erythroid 2-related factor 2 (NRF2) which is activated upon oxidative stress-induced degradation of its negative regulator kelch-associated protein 1 (KEAP1) [5]. Yet, in the presence of redox-active metals such as divalent iron (Fe^2+^) catalysing the Fenton reaction, hydroxyl radicals (HO∙) are generated from hydrogen peroxide (H_2_O_2_) [6]. Hence, limiting the availability of free divalent redox-active metals via sequestration within metal-binding proteins is an additional integral part of the cellular antioxidant defence machinery [7].

Paradoxically, despite OXPHOS being the most efficient way to generate ATP, many cancer cells have undergone metabolic reprogramming, wherein they mainly generate ATP from cytosolic aerobic glycolysis coupled to lactate fermentation. This metabolic re-programming in cancer was famously discovered by Warburg and Cori in the 1920s [8,9] and has been suggested as a cancer cell means to evade toxic levels of ROS production. However, maintenance of this Warburg effect requires higher glucose uptake and elevated metabolic activity making tumour cells nevertheless heavily reliant on the antioxidant machinery and maybe even more susceptible to oxidative stress [10,11]. Therefore, highly proliferative cancer cells are known to require handling of elevated cellular ROS levels in order to successfully establish tumours [12,13,14].

One of the most important hallmarks of cancer is the efficient evasion of regulated cell death. Recently, ferroptosis, a new, yet potentially evolutionary ancient type of regulated necrosis which is triggered upon collapse of a lipid radical-specific antioxidant defence system was described [15] (recently also reviewed [16,17,18,19,20]). Whereas apoptosis, necroptosis and pyroptosis are all either directly dependent on caspases or inhibited by their activity (reviewed in-depth elsewhere [21]), ferroptosis seems to have evolved separately with, as of yet, very little known direct molecular cross-talk to other pathways of regulated cell death [21]. As the study of ferroptosis is a relatively young, yet rapidly growing field, here, we will provide an updated systemic overview on processes regulating ferroptosis and potential outcomes in cancer.

## 2. Ferroptosis Pathway Regulation

One hallmark of ferroptosis is the requirement for iron, as demonstrated by the fact that chelation of iron by deferoxamine (DFO) rescues experimental induction of ferroptosis [15]. Consequently, transferrin, which binds free ferric iron (Fe^3+^) and shuttles it into cells, was shown to regulate ferroptosis [22]. Once Fe^3+^ is imported, endosomal six-transmembrane epithelial antigen of prostate 3 (STEAP3) catalyses the reduction to divalent iron (Fe^2+^) and releases it to the cellular labile iron pool through the divalent metal transporter 1 (DMT1) [23]. Interestingly, *DMT1* has been shown to be up-regulated upon ferroptosis-induction [24].

Another hallmark is the characteristic accumulation of membrane lipid peroxides preceding cell death [15]. Lipid peroxides were recently modelled to destabilise the lipid bilayer resulting in disintegration of cellular membranes in silico [25]. Through the use of lipidomics, arachidonic acid (AA)- and adrenic acid (AdA)-containing phosphatidylethanolamine (PE) species were identified as in vivo lipid products of ferroptosis [26]. These lipids can undergo spontaneous peroxidation in the presence of hydroxyl radicals (HO∙) generated from Fenton reactions of redox active divalent iron (Fe^2+^) and hydroperoxide (H_2_O_2_). Hydroxyl radicals (HO∙) can react directly with polyunsaturated fatty acids (PUFAs) in membrane phospholipids which can trigger a chain reaction of lipid ROS attacking proximal PUFAs. Alternatively, divalent iron can serve as a cofactor for lipoxygenase (LOX) to catalyse PUFA peroxidation enzymatically [27]. PUFAs are especially sensitive to lipid peroxidation due to the presence of highly reactive hydrogen atoms within methylene bridges [28]. Interestingly, 4-hydroxynonenal (4-HNE) and malondialdehyde (MDA) are fairly specific lipid peroxidation by-products, which have frequently been used as general markers of oxidative stress in tissue sections. Acyl-CoA synthetase long-chain family member 4 (ACSL4) mediates esterification of AA and AdA with coenzyme A (CoA) forming Acyl-CoA which can then undergo either ß-oxidation or anabolic PUFA biosynthesis [29,30,31]. Importantly, ACSL4 was identified to be required for cells to undergo ferroptosis by generating the lipid target pool for peroxidation [20,29]. In a similar manner, lysophosphatidylcholine acyltransferase 3 (LPCAT3) contributes to ferroptosis by incorporation of AA into phospholipids of cellular membranes thereby contributing to substrate generation for lipid peroxidation [29,32,33]. Together, these findings demonstrate that PUFA synthesis and peroxidation is an essential prerequisite for cells to die via ferroptosis.

Vice versa, GPX4 was shown to constitutively hydrolyse lipid hydroperoxides and thereby serve cellular protection from ferroptosis [34]. Antagonising GPX4 with the small molecule inhibitor rat sarcoma viral oncogene homolog (RAS)-selective lethal 3 (RSL3) led to efficient induction of ferroptosis [15]. GPX4 requires glutathione (GSH) as an electron donor to reduce lipid hydroperoxides. GSH is an abundant cellular tripeptide consisting of glycine, glutamate and cysteine and is utilised as one of the major cellular non-protein antioxidants [35]. GSH synthesis depends on the availability of intracellular cysteine which can be generated from cystine imported from the extracellular space via the sodium-independent cystine/glutamate antiporter System xc-. System xc- is a heterodimer consisting of a heavy chain (4F2, gene name *SLC3A2*) and a light chain (xCT, gene name *SLC7A11*) [36]. Interestingly, xCT, the subunit decisive for specific amino acid antiport was shown to be a molecular target of the small molecule eradicator of RAS and ST-expressing cells (erastin) and the resulting cystine depletion triggered ferroptosis [37,38].

Recently, ferroptosis suppressor protein 1 (FSP1), formerly called apoptosis-inducing factor mitochondria associated 2 (AIFM2), was identified as another ferroptosis protective factor in a CRISPR-Cas9 knockout screen for synthetic lethality with the GPX4 small molecule inhibitor RSL3 [39] and a cDNA overexpression screen complementing for *GPX4* loss [40]. Both studies reported that FSP1 is recruited to the plasma membrane by N-terminal myristoylation, where it acts as an oxidoreductase, reducing ubiquinone (=Coenzyme Q10) to the lipophilic radical scavenger ubiquinol which limits accumulation of lipid ROS within membranes in the absence of GPX4. Hence, ubiquinol generated by FSP1 acts as an endogenous functional equivalent of the described small-molecule lipophilic radical scavengers ferrostatin-1 (Fer-1) and liproxstatin-1 inhibiting ferroptosis [15]. Interestingly, in hundreds of cancer cell lines, *FSP1* expression correlated with ferroptosis resistance in non-haematopoietic cancer cell lines, yet most significantly in lung cancer cells, suggesting upregulation of FSP1 to be a strategy of ferroptosis escape in cancer [40,41].

## 3. Ferroptosis and Mitochondria

Mitochondria are indispensable for most normal cell types due to their role in generating ATP through OXPHOS [22,42]. However, this process comes at a cost of ROS production as a byproduct of OXPHOS [43]. Mitochondria are involved in the execution of various types of regulated cell death such as extrinsic and intrinsic apoptosis and autophagy, thereby playing a central role in tissue homeostasis [44,45]. Interestingly, experimental induction of ferroptosis through pharmacological inhibition of xCT was shown to induce mitochondrial fragmentation, mitochondrial ROS production, loss of the mitochondrial membrane potential (MMP) and ATP depletion [18,42,46,47,48,49]. Supporting a requirement for mitochondrial metabolism in the execution of ferroptosis [47], depletion of mitochondria via Parkin-mediated mitophagy in vitro or inhibition of OXPHOS rescued cells from ferroptosis induced by cystine deprivation or erastin [42]. Yet, in the initial characterisation of ferroptosis, mitochondrial DNA (mtDNA)-depleted ρ^0^ cells remained sensitive to oxidative stress and ferroptosis induction [15]. Hence, whether or not mitochondria are involved in ferroptosis across all cell types is still controversial and there may be cell-specific differences similar to a type I and type II cell scenario as described for extrinsic apoptosis [50]. Of note, the Bcl-2 family member BH3-interacting domain death agonist (BID), whose cleavage into truncated Bid (tBID) is known to be essential during extrinsic apoptosis in type II cells, was shown to be required for erastin-induced ferroptosis and oxytosis in neurons [46].

### 3.1. The Upstream and Downstream of Mitochondria in Ferroptosis

Interestingly, there seems to be a marked difference in the requirement for mitochondrial metabolism in the execution of ferroptosis depending on the strategy by which ferroptosis is triggered. When triggered by cystine starvation or by erastin, resulting in GSH depletion, activity of the mitochondrial TCA cycle was necessary for ferroptosis induction [22]. In fact, cancer cells deficient for the mitochondrial tumour suppressor fumarate hydratase (FH), a metabolic enzyme of the TCA cycle, were unable to undergo ferroptosis upon cystine deprivation [42]. Yet, when GPX4 was pharmacologically inhibited or deleted, cells underwent ferroptosis regardless of the TCA cycle, functional OXPHOS or mitochondria suggesting GPX4 activity required for ferroptosis prevention to lie downstream of mitochondria [42]. Supporting this idea, mitochondrial damage and mitochondrial ROS production are events taking place during the execution of ferroptosis upon inhibition of xCT or cystine starvation but are not necessary for GPX4 inhibition-induced ferroptosis [42,46]. Yet, arguing against an exclusive role of GPX4 downstream of mitochondria, apoptosis-inducing factor (AIF), which is associated with the inner mitochondrial membrane, translocates from mitochondria to the nucleus contributing to ferroptotic death upon *GPX4* deletion suggesting a certain level of mitochondrial permeability during ferroptosis [51]. This permeability, however, is Bcl-2 associated X protein (BAX) and Bcl-2 homologous antagonist/killer (BAK1) independent, as cells deficient for both proteins were able to undergo ferroptosis [15].

Interestingly, *GPX4* is expressed as three different splice variants before processing, which leads to initial sorting to different cellular compartments [52], whilst, after processing, these isoforms are identical. A signal peptide within the long isoform (lGPX4) leads to mitochondrial import, the short form (sGPX4) is found in the cytosol, mitochondria and microsomes and the nuclear isoform (nGPX4) in the nucleus [52]. Thereby, through these different localisations, GPX4 may fulfil compartment-specific functions which may account for the conflicting results obtained regarding mitochondrial involvement in ferroptosis after GPX4 deletion. Importantly, apart from its role in ferroptosis which has gained much attention recently, GPX4 was previously shown to preserve mitochondrial ATP generation by protecting mitochondria from ROS build-up during tert-Butyl hydroperoxide (TBHP)-induced cell death [53]. Taken together, GPX4, or at least a particular isoform thereof, seems to exert an important function in the prevention of ferroptosis independently of mitochondria, and its role in protecting mitochondrial metabolism from ROS-induced membrane damage may also feed into its ferroptosis-protective activity (summarised in Figure 1).

### 3.2. Lipid Peroxidation in Mitochondria

GPX4 is one of the major intracellular enzymes involved in hydrolysing lipid peroxides, thereby ensuring repair of lipid peroxide-perturbed cellular membranes. In a counter regulation step, GPX4 inhibition and GSH depletion induces the activation of 12/15 lipoxygenases (12/15-LOX), which are involved in lipid peroxidation [51,54]. Once 12/15-LOX are activated they can directly attack the mitochondrial membrane inducing local lipid peroxidation in neurons [54,55,56]. Therefore, although the current view of ferroptosis favours the idea of lipid peroxide accumulation in the plasma membrane leading to cellular rupture [25], lipid peroxides have also been shown to accumulate in the mitochondrial membrane during ferroptosis. In addition to 12/15-LOX regulating this process, CDGSH iron sulfur domain 1 (CISD1) is an iron-containing protein whose N-terminus is inserted into the outer mitochondrial membrane where it regulates mitochondrial iron uptake. Upon CISD1 deletion, iron accumulation inside the mitochondria facilitates the generation of mitochondrial lipid peroxides contributing to ferroptosis [31]. On the other hand, cholesterol hydroperoxide (ChOOH) species (5alpha-OOH, 6alpha/6beta-OOH, 7alpha/7beta-OOH) and phospholipid hydroperoxide (PLOOH) families (PCOOH, PEOOH, PSOOH) which are peroxidised in the cytosol can be transported to mitochondria via the sterol carrier protein 2 (SCP-2) upon cystine deprivation [55,57]. This suggests a potential role for these lipid peroxides in mitochondria during ferroptosis triggered via this route.

Taken together, mitochondrial metabolism is a main source of cellular ROS and contributes to ferroptosis in many cellular systems. However, mechanisms of how ferroptosis and mitochondria cross signal and whether xCT and GPX4 always signal in a single hierarchical pathway is poorly understood. Therefore, additional studies on the mechanistic contribution of mitochondria in ferroptosis are warranted.

## 4. Ferroptosis Discovery as an Oncogene-Selective Death

Although not named at the time, ferroptosis was first discovered as part of a synthetic lethality screen for small molecules selectively targeting Harvey rat sarcoma viral oncogene homolog *(HRAS)^G12V^*-mutant human foreskin fibroblasts (BJeLR) [58]. Next, the small molecules erastin [48] and RSL3 [59] were described to induce an oncogenic RAS-specific oxidative type of cell death independently of caspases. The observed *HRAS^G12V^* selectivity was proposed to stem from an increased expression of transferrin receptor *TFRC* and thereby increased intracellular iron levels in *HRAS*-mutant cells [59]. Moreover, Kirsten sarcoma viral oncogene homolog *(KRAS)*-mutant Calu-1 lung cancer cells exhibited higher sensitivity to erastin and silencing of *KRAS* by small hairpin (sh) RNA reduced erastin’s efficacy in these cells. Additionally, A-673 cells, which harbour an activating *BRAF^V600E^* mutation, became more resistant to erastin treatment upon shRNA-mediated suppression of *BRAF* [48]. Supporting an increased ferroptosis sensitivity of oncogene-expressing cells, human mammary epithelial *(HME)* cells expressing mutant epithelial growth factor receptor *(EGFR)* were more sensitive to cystine deprivation-induced ferroptosis through increased mitogen-activated protein kinase (MAPK) pathway activation [60]. Additionally, in non-small-cell lung cancer (NSCLC) cell lines, it was shown that the level of MAPK pathway activity correlates with sensitivity to ferroptosis induced by cystine deprivation [60].

Apart from their potential influence on ferroptosis execution, oncogenic *RAS* isoforms and induction of cellular ROS have been extensively studied. In this regards, oncogenic *NRAS^G12D^* and *HRAS^G12V^* expression was shown to activate ROS production and to trigger a p38 mitogen-activated protein kinase (MAPK)-mediated oxidative stress response [61]. Moreover, RAS-stimulated ROS production was shown to be mediated via the activation of nicotinamide adenine dinucleotide phosphate hydrogen (NADPH)-oxidases (NOX) regulated by the PI3K/Rac1 and RAF/MEK/ERK RAS-effector pathways [62,63]. KRAS also induced translocation of p47^phox^, a subunit of NADPH oxidase 1 (NOX1), to the plasma membrane thereby promoting its activation and aiding cellular transformation [64]. Additionally, in the context of inactivation of the tumour suppressor *p16*, *KRAS^G12V^* expression also up-regulated NADPH oxidase 4 (NOX4) [65]. Thereby, oncogenic *RAS* isoforms may also influence lipid peroxidation and ferroptosis through enhanced feeding into the general cellular ROS pool.

Whilst these and other studies have demonstrated an induction of ROS under acute oncogenic *RAS* overexpression in vitro, which may promote cellular transformation at the cost of elevated ROS, the question remained how tumours expressing oncogenic *KRAS* from the endogenous locus would handle ROS in vivo. Interestingly, expression of *KRAS^G12D^*, *BRAF^V619E^* and *MYC^ERT2^* oncogenes from their respective endogenous loci activated nuclear NRF2, a major transcription factor inducing antioxidant defence genes [66,67]. Importantly, NRF2 is responsible for the positive regulation of different genes involved in GSH synthesis including *xCT*, glutamate-cysteine ligase catalytic subunit *(GCLC)* and glutamate-cysteine ligase modifier subunit *(GCLM)* [68,69,70]. In addition, NRF2 promotes expression of Ferritin (FTH) [71] which may act as a scavenger for redox active iron suggesting a protective function of NRF2 in ferroptosis. In line with this suggestion, hepatocellular carcinoma cells became more sensitive to ferroptosis inducers upon deletion or pharmacological inhibition of NRF2 [71,72]. NRF2 interacts with KEAP1, a tumour suppressor protein, which also regulates the expression of the ATP binding cassette (ABC)-family transporter multidrug resistance protein 1 (MRP1). Interestingly, MRP1 was shown to sensitise to ferroptosis by mediating glutathione efflux [73]. Additionally, it was shown that there is a high co-occurrence of *KEAP1* and *KRAS* mutations in human lung cancers which elevates cellular rates of glutaminolysis [74]. Increased glutamate shuttling into the TCA cycle, namely glutaminolysis, may compete with glutamate requirement for cystine antiport which might affect GSH levels. Yet, NRF2 was also shown to be activated by withaferin A leading to induction of its bona-fide target gene heme oxygenase 1 *(HMOX1)* which causes an excess of cytosolic labile iron through catalysing its release from haeme promoting ferroptosis in neuroblastoma [75]. Hence, NRF2 activation and target gene expression can lead to opposing outcomes for ferroptosis and its actual effect on ferroptosis is possibly cell type specific.

Also arguing against mutant *KRAS* as a marker of ferroptosis sensitivity, artesunate (ART) was shown to induce ferroptosis in an iron- and ROS-dependent manner in pancreatic ductal adenocarcinoma (PDAC) cell lines irrespective of *KRAS* status [76]. Furthermore, erastin treatment induced growth inhibition in acute myeloid leukaemia (AML) cells with an *NRAS^Q61L^* mutation (HL-60), but not in other cell lines harbouring *RAS* mutations (e.g., *NRAS^G12D^* or *KRAS^A18D^*) or *RAS* wild type (WT) suggesting other genetic or non-genetic factors influencing ferroptosis sensitivity [77]. Moreover, there is evidence suggesting that oncogenic *KRAS* mutations may in fact protect cells from ferroptosis. In this regards it was shown that rhabdomyosarcoma cells (RMS13) expressing either *NRAS^G12V^*, *HRAS^G12V^* or *KRAS^G12V^* were more resistant to ferroptosis than respective empty vector control cells [78]. Interestingly it was recently shown that fibroblasts, expressing oncogenic *KRAS^G12V^*, are protected from hydrogen peroxide-induced cell death by up-regulating xCT which allows for KRAS-induced tumourigenicity in vivo [79].

It was demonstrated that the mitochondrial TCA cycle is required for ferroptosis [42]. Interestingly, a shift in the metabolic pathway in *KRAS*-driven cancers can enhance different characteristics like glutaminolysis, glycolysis or nutrient uptake and thereby affect the TCA cycle [80]. Moreover, NADPH is needed to maintain the TCA cycle, fatty acid synthesis and glutamine metabolism and is used by glutathione reductase (GR) to reduce oxidised GSSG to GSH. Of note, KRAS was shown to elevate NADPH levels through metabolic reprogramming which may enable an improved rate of GSH regeneration and ferroptosis protection [81,82].

Intriguingly, cells expressing homozygous *KRAS^G12D/G12D^* demonstrated upregulation of genes related to glycolysis, enhanced glucose consumption and significantly increased levels of TCA cycle enzymes. Moreover, it was shown that homozygous KRAS mutant cells shuttle glucose towards glutathione synthesis, again suggesting that these cells may be more protected from ROS and thereby more resistant to ferroptosis [83].

Apart from oncogene status, in a panel of 117 cancer cell lines from different tissues, sensitivity to erastin-induced cell death was examined to define possible additional determinants of erastin sensitivity. Interestingly, tissue origin was a much stronger predictor of ferroptosis sensitivity in cancer cell lines than oncogene mutational status [34]. Within this study, diffuse large B-cell lymphoma (DLBCL) cell lines were identified as particularly sensitive to ferroptosis [34]. Although, it was described that suspension cells are more sensitive to small molecules which induce growth inhibition, this was not the underlying mechanism for increased ferroptosis sensitivity in DLBCL which remains to be addressed [34,84]. Apart from tissue origin, another important factor determining ferroptosis sensitivity may be cellular differentiation. Interestingly, in melanoma, ferroptosis sensitivity did not correlate with MAPK pathway activity but instead depended on the dedifferentiation status [34]. Moreover, cells with an expression signature indicative of an epithelial-to-mesenchymal transition (EMT) were more sensitive to GPX4 inhibitors [85]. Interestingly, high cell confluence inhibits ferroptosis via loss of YAP-mediated transcriptional upregulation of *TFRC* and *ACSL4*, an experimental variable which therefore needs to be tightly controlled in studies aiming for the identification of factors influencing ferroptosis sensitivity [86].

## 5. In Vivo Relevance of Ferroptosis: Lessons Learned from Knockout Mice

One of the key bottlenecks for protection from ferroptosis is the availability of GSH, which serves as a redox equivalent for GPXs including GPX4. As mentioned above, one important source of intracellular cysteine for GSH synthesis is cystine imported via xCT (=*Slc7a11*). xCT is highly expressed in neurons and brush border membranes of the kidney and duodenum [37]. Furthermore, it has been reported to be expressed in the thyroid gland [87]. Expression can be induced by oxidative stress stimuli such as hydrogen peroxide, oxygen and sodium arsenite leading to elevated GSH production [87,88]. In line with a role in cystine import, xCT-deficient mice display increased **cystine** concentrations in blood plasma and decreased intracellular GSH levels in comparison to WT mice. However, cellular cysteine levels are comparable between knockout (KO) and WT littermates, suggesting compensatory cysteine synthesis via the transsulfuration pathway [87]. Mouse embryonic fibroblasts (MEFs) isolated from *Slc7a11* KO mice fail to survive in cell culture except when supplemented with 2-mercaptoethanol or N-acetyl cysteine (NAC) both described to serve as alternative cystine sources for cells. These data suggest that cells grown in 2D culture in vitro may lack compensation via the transsulfuration pathway. Interestingly, *Slc7a11* KO mice are protected from neurotoxic insults induced by 6-hydroxydopamine (6-OHDA), which can trigger Parkinson’s Disease (PD) via decreasing extracellular glutamate levels in the brain [89].

During GSH synthesis, glutamate and cysteine are ligated by the glutamate-cysteine ligase (GCL) forming the GSH precursor γ-glutamylcysteine (see Figure 1). GCL is a heterodimer consisting of the catalytic subunit GCLC and the regulatory subunit GCLM [90]. Deletion of *Gclc* is embryonically lethal before E8.5 as mice are unable to synthesise GSH [91,92]. Moreover, *Gclc* knockout embryos fail to gastrulate and instead present with increased cell death assayed by TUNEL staining [92]. In contrast, *Gclm*-deficient mice develop normally but synthesise approximately 75–90% less GSH in comparison to WT littermates, suggesting that either 10–25% of cellular GSH are sufficient for healthy embryonic development or *Gclc* may fulfil additional functions during embryonic development [93]. Glutathione synthetase (GSS) catalyses condensation of glycine and γ-glutamylcysteine to generate GSH. Mice deficient in *Gss* die at E7.5, whilst heterozygous littermates survive and do not display an overt phenotype [94]. These data underline the important role of GSH synthesis for normal embryogenesis and development and may in the case of *Gclc*-deficiency hint at a connection with cell death prevention during embryonic development.

The transsulfuration pathway represents an alternative strategy for cysteine generation from methionine, in which the cystathionine beta-synthase (CBS) processes homocysteine [95,96]. *Cbs* deletion in mice results in severe growth retardation and postnatal death 5 weeks after birth. This is accompanied by elevated levels of the cysteine precursor homocysteine in plasma, generating a mouse model for severe homocystinuria [97,98]. Interestingly, a recently developed CBS inhibitor has been identified to induce ferroptosis in different cancer cell lines [99]. Yet, whether any aspect of the *Cbs* KO phenotype is related to overt induction of ferroptosis is unknown.

Downstream of GSH synthesis, GPX4 is a GSH-dependent key regulator of the ferroptosis pathway. Human GPX4 contains a selenocysteine encoded by a “Stop” codon within its catalytic domain. GPX4 has been shown to be indispensable for embryogenesis as *Gpx4* KO mice die in utero at E7.5 and display abnormal organ compartmentalisation [100,101]. Interestingly, mice with an inactivating serine exchange mutation in the enzymatically active selenocysteine of Gpx4 die at the same stage of embryonic development whereas mice with heterozygous loss of selenocysteine within Gpx4 are born but display defects in spermatogenesis [52,102]. Of note, mice containing a cysteine instead of a selenocysteine in the catalytically active site of Gpx4 are viable but display seizures and cell death in interneurons when on a mixed genetic background [103,104]. Whole-body inducible knockout of *Gpx4* increased oxidative stress and mitochondrial dysfunction in *Gpx4*-depleted organs with decreased activity of electron transport chain members complex I and IV demonstrating that *Gpx4* is an important protector of mitochondrial integrity [105]. Adult mice die within two weeks after systemic induction of *Gpx4* KO caused by acute renal failure demonstrating its essential role also for adult tissue homeostasis [55,105]. Interestingly, *Gpx4* deletion in hematopoietic cells causes receptor-interacting protein 3 (*Rip3*)-dependent cell death in erythroid precursor cells resulting in anaemia in mice [106]. Mechanistically, the authors show that GPX4 deletion leads to glutathionylation and inactivation of caspase 8, which triggers necroptosis independently of tumor necrosis factor (TNF). Moreover, T-cell-specific deletion of *Gpx4* resulted in T cell ferroptosis which was dependent on receptor-interacting protein 1 (*Rip1*) and *Rip3*, suggesting that Gpx4 may be essential for T-cell mediated immune responses [107]. Thereby, these two studies strongly suggest an interaction between ferroptosis and necroptosis takes place in vivo. This has also recently been discussed in Florean et al. [19].

Mice harbouring a x-chromosomal deletion of the important PUFA-regulator *Acsl4* are viable but adipocyte-specific deletion results in decreased levels of PUFA-derived fatty acyl-CoAs which fuels lipid peroxidation [108,109].

Thereby, protection from ferroptosis plays a crucial role for tissue homeostasis, whilst many proteins involved in ferroptosis also fulfil metabolic functions unrelated to regulated cell death which may be causative for some of the phenotypes seen in KO mouse models.

### Cancer Mouse Models and Genetic Evidence for Ferroptosis in Cancer

In mouse models of mammary tumours, lymphomas and sarcomas, *Gclm* deficiency leading to a significant reduction of cellular GSH caused impaired tumour initiation and progression [110]. Moreover, murine MC38 colon cancer and Pan02 pancreatic cancer cells, harbouring a CRISPR-Cas9-mediated *Slc7a11* knockout displayed impaired in vivo tumour growth accompanied by increased ROS levels and decreased GSH levels [111]. Interestingly, Arensman et al. showed that while cystine uptake by Slc7a11 is essential for T cell viability and proliferation in vitro, it is dispensable for T cells in vivo and does not impact T cell infiltration in tumours in immunocompetent mice. These results show that targeting xCT unlike GPX4 [107] may represent an effective cancer treatment strategy without compromising anti-tumour immunity. Recently, it has been demonstrated that *Gpx4* KO significantly impairs tumour growth upon Fer-1 withdrawal, whilst additional *Fsp1* KO enhances this effect [39].

## 6. Ferroptosis as a p53-Mediated Tumour-Suppressive Mechanism

Cell cycle arrest, cell death and senescence can, in many cases, serve tumour suppression. Therefore, p53, a master regulator of cell cycle arrest and transcription of intrinsic apoptosis genes, has long been assumed to exclusively suppress tumours via these mechanisms [112,113,114]. Surprisingly, mutating three lysine residues within the DNA-binding domain of p53 (*p53^3KR^*), which prevents their acetylation, was sufficient to suppress tumour growth in mice despite impaired functionality of this mutant in inducing cell cycle arrest, senescence and apoptosis [115]. Strikingly, it was discovered that the *p53^3KR^* mutant was still capable of suppressing xCT transcription via direct promotor binding, thereby sensitising cells to ferroptosis upon ROS-induced stress [116]. Hence, ferroptosis induction has been identified to belong to the arsenal of tumour-suppressive activities of p53. Importantly, mutating a fourth identified acetylation site in *p53* (K98) in addition to the three previously described ones led to a complete loss of its tumour suppressor activity. Importantly, this mutant had also lost the capacity to induce ferroptosis sensitisation through xCT suppression, which indicates that the acetylation of the fourth acetylated lysine residue K98 in *p53* in addition to the other three sites is vital for tumour suppression and ferroptosis induction by *p53* [117]. Interestingly, cytosolic accumulation of *p53* mutants typically found in cancer mutants was shown to bind and thereby sequester NRF2, preventing nuclear translocation of NRF2 and induction of its target genes including xCT [118]. These data might offer an explanation as to how p53 mutants may indirectly promote suppression of xCT expression irrespective of DNA binding. In addition, cytoplasmic accumulation preceding nuclear translocation may contribute to wild type p53-mediated xCT suppression. Another study demonstrated that the INK4 locus alternative reading frame (ARF) protein functions as a p53-independent tumour suppressor by limiting NRF2-mediated xCT induction resulting in tumour growth suppression [68]. Thereby, ferroptosis sensitivity is also linked to ARF status and its interaction with NRF2. Interestingly, APR-246, a clinical reactivator of mutant p53 was shown to decrease intracellular glutathione levels through direct binding of cysteines within GSH, leading to an increase of cellular ROS [119]. Yet, it remains to be established whether APR-246-mediated decrease in cellular GSH also stems from p53-mediated xCT suppression and resulting decrease of cystine import. Interestingly, a p53 Ser47 single nucleotide polymorphism (SNP) was identified in a population of African descent which rendered cells more resistant to RSL3 treatment and generated knock-in mice more prone to spontaneous tumour development [120]. This Ser47 variant impaired phosphorylation on adjacent Ser46, which was important for p53 to induce spontaneous cell death in different cell lines [121]. Moreover, a lack of this phosphorylation in Ser46 decreased the ability of Ser47 to bind to p53 target genes which might explain decreased RSL3 sensitivity by de-repression of xCT [120].

In contrast to the above-mentioned studies showing that p53 can induce ferroptosis through suppression of xCT expression [116,118], in human colorectal cancer (CRC) cells, p53 was shown to promote SLC7A11 expression [122]. Furthermore, the same study showed that loss of p53 inhibits accumulation of dipeptidyl-peptidase-4 (DPP4) in the nucleus, which results in enhanced plasma-membrane associated DPP4-dependent lipid peroxidation via ROS-generating NOX enzymes resulting in ferroptosis [122]. Thereby, loss of p53 may equally sensitise cells to ferroptosis in certain contexts. In addition, treatment with the MDM2 inhibitor nutlin-3 stabilised WT p53, reduced cellular ferroptosis sensitivity and induced p21 [123]. p21 upregulation in turn promoted intracellular glutathione storage leading to reduced accumulation of lipid ROS also in the presence of p53 transcriptional activity [123].

Although many of these studies propose that there is a relationship between tumour suppression and ferroptosis sensitivity, so far, there is no genetic evidence that p53 expression directly induces ferroptosis and thereby mediates tumour suppression in vivo. Moreover, results obtained with loss of p53 do not account for the NRF2 sequestering function seen for mutated p53 in the cytosol. Therefore, future studies will have to unravel whether ferroptosis can suppress tumour growth and to what extent it is part of the constitutive p53-mediated tumour-suppressive machinery.

## 7. Ferroptosis-Inducing Therapy (FIT) for Cancer Treatment

A breakthrough discovery for a potential use of ferroptosis-inducing therapy (FIT) for cancer treatment came with the finding that cells with acquired resistance to the human epidermal growth factor receptor (HER1/HER2) tyrosinkinase inhibitor Lapatinib, so called persister cells with a high mesenchymal state, were selectively sensitised to the induction of ferroptosis [85,124]. These studies for the first time pointed out that cells which had escaped other means of killing may be selectively sensitised to ferroptosis. Moreover, immunotherapy was recently shown to sensitise to tumour cell ferroptosis (see Section 9). Considering that xCT is often aberrantly expressed in many cancers [125,126], ferroptosis induction may just prove to be a weak spot of cancer.

However, the two main ferroptosis inducers used in vitro, RSL3 and erastin, do not meet pharmacokinetic standards for in vivo application yet due to poor water solubility and metabolic instability [34,127,128]. To circumvent this problem, several efforts have been undertaken to render erastin more suitable for in vivo application. In one approach, triple-negative breast cancer (TNBC) cells, which highly express folate receptor, were treated with erastin packaged in exosomes covered with folate to specifically target folate receptor-overexpressing TNBC [127]. Another metabolically more stable form of erastin is piperazine-coupled erastin which has demonstrated anti-tumour activity in a xenograft model using human fibrosarcoma HT-1080 cells [15,34]. In addition, imidazole-ketone erastin, a metabolically stable variant of erastin, was shown to reduce tumour growth in a SU-DHL-6 DLBCL xenograft model [128].

Whilst small molecule backbones for RSL3 and erastin will have to be further optimised for clinical application, interestingly, a number of food and drug administration (FDA)-approved drugs have been identified to function via the induction of ferroptosis in different cancer entities:

Sorafenib, is an FDA-approved multi-kinase inhibitor for treatment of advanced renal cell carcinoma (RCC) and advanced hepatocellular carcinoma (HCC). Molecularly, Sorafenib was shown to inhibit system xc- [38]. Moreover, cell death induced in HCC by sorafenib was suppressed by ferropstatin-1 and iron-chelators [129]. It has further been reported that the tumour suppressor retinoblastoma protein (RB1) suppresses ferroptosis induced by sorafenib treatment [130] suggesting a possible biomarker for sorafenib treatment-induced ferroptosis.

Sulfasalazine (SAS), an FDA-approved drug for the treatment of rheumatoid arthritis and inflammatory bowel diseases (Crohn, ulcerative colitis), is thought to act as anti-inflammatory drug [131]. Described targets of SAS are arachidonat-5-lipoxygenase (ALOX-5) [132], cyclooxygenase 2 (COX-2) [133] and nuclear factor ’kappa-light-chain-enhancer’ (NF-κB) [134]. It has also been shown to inhibit the system xc- subunit xCT and effectively induce ferroptosis in non-Hodgkin lymphoma cells [131].

Altretamine (hexamethylmelamine), an FDA-approved alkylating antineoplastic drug, is used for the treatment of ovarian cancer [135]. It has also been shown to inhibit GPX4 and effectively kill U-2932 DLBCL cells in vitro [136].

Statins, such as cerivastatin and simvastatin, have been shown to reduce the synthesis of Coenzyme Q10 via blocking the mevalonate pathway and thereby induce ferroptosis in the human fibrosarcoma cell line HT-1080 [85].

Thereby, the induction of ferroptosis may underlie treatment efficacies of several already approved cancer drugs which may shorten clinical development for the concept of therapeutic induction of ferroptosis in human cancers [17,19]. Which ones these are apart from the already identified ones above will have to be discovered in the future. Current compounds known to induce ferroptosis are summarised in Table 1.

## 8. Ferroptosis and Inflammation in Pathology

A hallmark of regulated necrosis pathways is the loss of plasma membrane integrity [143]. This distinct feature of regulated necrosis allows for the release of cytosolic content and exposure of this content to the surrounding tissue. Many of these released cytosolic factors function as immunogenic signals, so called damage-associated molecular patterns (DAMPs) [144]. Not only DAMPs but also alarmins that are released upon cell death or injury have the ability to induce inflammatory immune responses [145]. Mechanistically, DAMPs and alarmins operate as ligands that stimulate pattern recognition receptor (PRR)-expressing immune cells resulting in an inflammatory process, termed necroinflammation if regulated necrosis was the initiator of immune activation [144]. Although precise mechanisms remain to be established, accumulating evidence suggests that ferroptosis may be inflammatory through lipid peroxidation-mediated plasma membrane rupture [146] leading to sterile inflammation [36]. As such, the small molecule inhibitor Fer-1 reduced immune cell infiltrations into diseased tissues in models of acute kidney injury (AKI) [147,148]. Fer-1 treatment also decreased cytokine and chemokine expression levels (C-X-C-motif chemokine 2 (CXCL-2), interleukin 6 (IL-6), p65 subunit of NF-κB [147], interleukin 33 (IL-33), TNF-α, monocyte chemotactic protein 1 (MCP-1) [148]), suggesting that ferroptotic DAMPs may have the capacity to elicit secondary immune cell activation and cytokine production or that production of these alarmins is directly blocked by Fer-1. Interestingly, the application of the caspase inhibitor zVAD-fmk and the RIPK1 inhibitor necrostatin-1 appeared to be futile for folic acid-induced AKI amelioration [148]. Mononuclear interstitial infiltrations were also observed to occur upon *GPX4* deletion in kidneys [55]. These data suggest that inflammatory processes and activation of the immune system in acute kidney injury are induced through ferroptotic cell death.

Fer-1 treatment further prevented neutrophil recruitment and their interferon (IFN)-dependent adhesion to coronary veins following ischemia/reperfusion injury (IRI) through a Toll-like receptor 4 (TLR4)-mediated signalling pathway [149]. Ferroptotic cell death was also shown to contribute to human diseases affecting the brain such as intracerebral haemorrhage [150] as well as neurodegeneration [151,152]. Interestingly, conditional neuron-specific *Gpx4* knockout mouse models revealed up-regulated proinflammatory cytokine levels (TNF-α and IL-6) associated with neuronal degeneration and cognitive impairment [151].

Furthermore, Trolox, another radical scavenger known to block ferroptosis, was equally shown to decrease expression levels of proinflammatory cytokines (TNF-α, IL-6, interleukin 1 beta (IL-1β)) in steatohepatitis [153]. This metabolic liver disease is characterised by the progression from steatosis into fibrosis through inflammatory processes [154]. Overall, it is strongly suggested that ferroptosis is closely associated with overt inflammatory signatures exhibiting elevated levels of proinflammatory cytokines in damaged tissues [147,148,151,153]. Collectively, these results imply that ferroptosis exerts a critical role in early inflammatory processes and that the prevention of necroinflammation provides a novel therapeutic strategy in combating inflammatory tissue damage and disease.

Although other regulated necrosis pathways such as necroptosis and pyroptosis were shown to mitigate inflammation in several diseases [155], a growing body of literature suggests that ferroptosis may be present from the outset of inflammatory [20,148,149,153] potentially triggering or sensitising to other inflammatory events. Importantly, immunogenic cell death results in the recruitment and activation of immune cells through alarmins which in turn has been hypothesised to induce further necrotic cell death pathways, culminating in a necroinflammatory auto-amplification loop [156]. However, the exact molecular pathways/machineries driving primary and secondary necroinflammation and the underlying hierarchical sequence of events remains yet to be disentangled.

### M1 versus M2-Type Immunity in Cancer: Potential Implications for Ferroptosis

Necroptotic cancer cells can release DAMPs, triggering inflammation in normal tissues [157]. Recently, it was shown that ferroptosis can equally allow for the release of the known DAMP high-mobility group protein B1 (HMGB1) in a manner dependent on the cellular autophagy machinery [158]. Apart from this, it is conceivable that the induction of ferroptosis in cancer cells might elicit innate immune responses which can influence tumours via various ways. The M1/M2 polarisation axis proposed by Mills et al. in 2000 provides a simplistic approach to distinguish between two types of macrophages with different metabolic programmes [159]. Whereas the M1 macrophage phenotype was reported to result in tumour regression [160] and inhibition of tumour growth [161], M2 macrophages have been shown to exhibit tumour-promoting activity as they produce angiogenesis factors [162] and inhibit M1 macrophages, thereby suppressing anti-tumour immunity. The importance of macrophage polarisation in the tumour microenvironment was further demonstrated by showing a chemokine-mediated decrease in M2-like macrophages in the vicinity of the tumour leading to reduced tumour growth [163]. In view of the intricate dynamics between innate and adaptive immune responses, it is important to highlight that M1/M2 macrophages promote adaptive Th1/Th2 lymphocyte responses, respectively [159]. M1/Th1 responses are associated with the release of proinflammatory cytokines, such as TNF [164] as well as IFN-γ, a cytokine that is in turn able to stimulate M1 polarisation in a feed-forward loop [165] as well as downregulate xCT [137]. In contrast, macrophage-induced Th2 responses are closely linked to anti-inflammatory mediators, such as IL-4 and IL-10 [164,166]. Interestingly, cytokines released from Th2 T-cell clones have been shown to avert cytokine production by Th1 cells, essentially strengthening the suppression of M1/kill type activity [159,167]. Furthermore, tumour cells have developed mechanisms to stimulate tumour cell migration and to maintain a high M2 to M1 ratio in the tumour microenvironment, thereby working in their best interest [168,169,170]. Of note, the majority of tumour-associated macrophages (TAM) in the tumour microenvironment is predominantly composed of M2 macrophages [171]. As a result, anti-cancer macrophage innate conversion therapies have emerged in recent years, directly targeting intracellular signalling pathways to increase the M1/M2 ratio [172,173]. Beatty et al. showed that the combination therapy of gemcitabine and agonist CD40 antibody obtained macrophage-dependent tumour regression in vivo which was suggested to be attributable to M1 activity [174].

As the M1/M2 dichotomy influences inflammatory events in opposite directions, it is of great interest to elucidate the quality of a ferroptotic secretome released from dying cancer cells as it may either trigger M2 tumour-promoting activity or initiate an M1 anti-tumour immune response. Given that ferroptosis in tissues has been causatively linked to the presence of TNF, this may suggest the potential for the promotion of an M1-type immune microenvironment. Supporting the idea of ferroptosis to be inflammatory in the context of cancer, ferroptosis triggered in neuroblastoma by withaferin A led to an influx of immune cells into the tumour [75]. Moreover, recent research has proposed that sensitising tumour cells to ferroptosis enhances the efficacy of combinatorial cancer immune-activatory therapies [175]. Interestingly, xCT was reported to be down-regulated through both immunotherapy-activated IFNγ [137,175] as well as radiotherapy-induced ataxia-telangiectasia mutated gene (ATM), resulting in increased tumour cell lipid peroxidation and tumoural ferroptosis [175]. Furthermore, immunotherapy-induced cytokines, namely IFNγ and TNF-α, were shown to provoke dedifferentiation in melanoma cells which was accompanied by resistance development. Intriguingly, this transition by tumour cells resulted in an increased susceptibility to ferroptosis induction and affected their overall immunosuppressive abilities [176]. Therefore, not only the induction but the sensitisation of tumour cells to ferroptosis may supply a novel tool for anti-tumour immunity and may serve the optimisation of established immunotherapies.

## 9. Conclusions

Many cancers highly express xCT, suggesting a selective dependency on either cystine or GSH, which may be exploited therapeutically by targeting these signalling nodes. Moreover, cancer cells appear to acquire ferroptosis sensitivity as part of an escape strategy against other targeted therapies, posing an opportunity for FIT in therapeutic management of relapse. Whilst the pathway will require additional in-depth characterisation and the validity of genetic induction of ferroptosis remains to be tested in genetically engineered mouse models of cancer, intriguingly, immunotherapy seems to positively interact with ferroptosis in vivo. These and other data strongly suggest ferroptosis to either be directly immunogenic or to prime an inflammatory response to other forms of regulated necrosis in the tumour microenvironment. Collectively, we propose that tumour cell ferroptosis may promote four possible outcomes in cancer: (A) if immuno-silent and completely killing, tumour cell ferroptosis should suppress tumours, (B) if immuno-silent but fractional killing causes selection, ferroptosis may result in tumour promotion through selection of the fittest cancer cell clone, (C) if an M1-type immunity is triggered, ferroptosis should be tumour-suppressive via activation of anti-tumour immunity and (D) if an M2-type immunity is triggered, ferroptosis would be overall tumour-protective, as it would aid shielding tumours against anti-tumour immune attack (proposed concepts summarised in Figure 2). Importantly, mixed responses within these four response types are likely to occur due to tumour heterogeneity. Thus, it will be important in future work to determine not only cellular determinants of ferroptosis sensitivity and resistance but also systemic responses and mechanisms of how these are interlinked with other types of regulated cell death in order to fully harness the potential of FIT for cancer treatment.

## Figures and Tables

**Figure 1 cancers-12-00164-f001:**
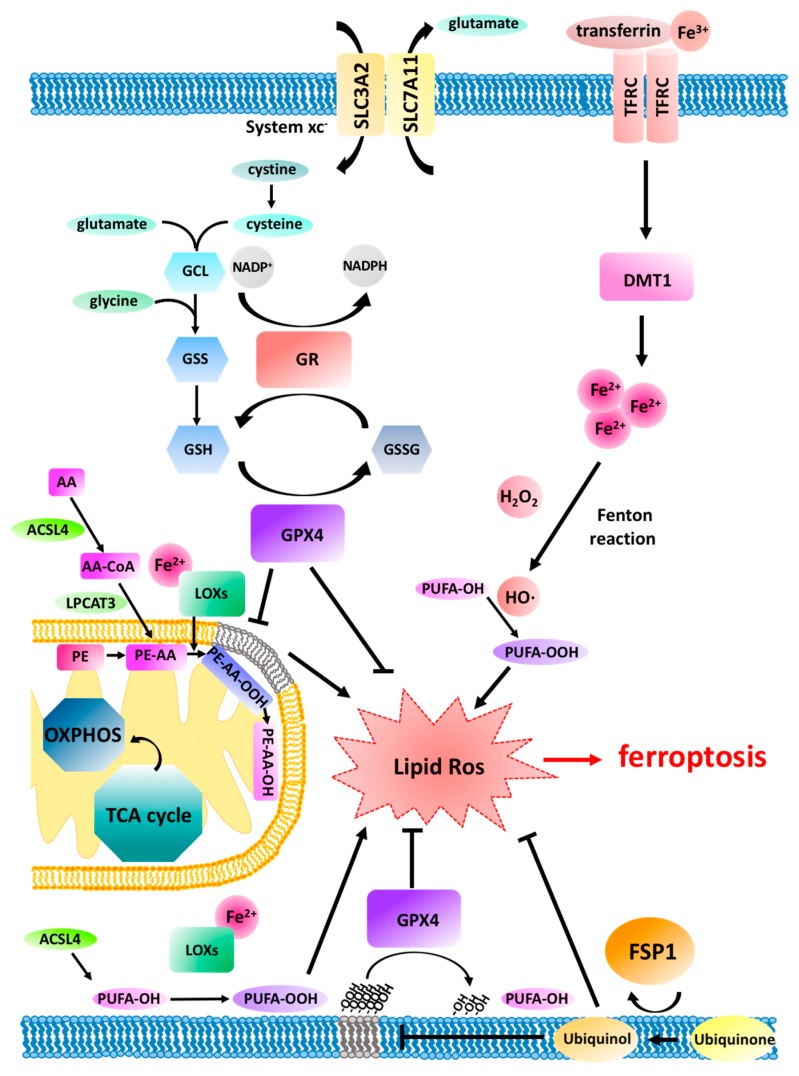
Schematic view of the ferroptosis pathway. Ferroptosis pursues upon aberrant build-up of lipid reactive oxygen species (ROS) leading to peroxidation (-OOH) of polyunsaturated fatty acids (PUFAs). Main peroxidation target PUFAs are arachidonic acid (AA) phosphatidylethanolamine (PE) lipid species within cellular membranes leading to membrane destabilisation and rupture. Lipid peroxidation can be triggered by cytosolic redox active iron (Fe^2+^) shuttled into cells bound to transferrin via transferrin receptor (TFRC) endocytosis and endosomal release mediated by divalent metal transporter 1 (DMT1). In the presence of H_2_O_2_, Fe^2+^ catalyses hydroxyl radical (HO∙) generation in a Fenton reaction, which sets of a radical lipid peroxidation chain reaction. Lipoxygenase (LOX) can equally catalyse lipid peroxidation using Fe^2+^. As a required prerequisite for ferroptosis, Acyl-CoA synthetase long-chain family member 4 (ACSL4) and lysophosphatidylcholine acyltransferase 3 (LPCAT3) generate the pool of AA-containing target lipids. Glutathione peroxidase 4 (GPX4), in turn, hydrolyses lipid peroxides converting them into their respective non-toxic lipid alcohols (-OH). GPX4 requires glutathione (GSH) as a cofactor which upon its oxidation (GSSG) by GPX4 is reduced to GSH by glutathione reductase (GR). GSH synthesis depends on glutamate cysteine ligase (GCL) and glutathione synthetase (GSS) as well as on intracellular cystine shuttled into the cell in exchange for glutamate mediated by system xc^-^ (SLC3A2 and SCL7A11/xCT). Independently of GSH, ferroptosis suppressor protein 1 (FSP1) generates ubiquinol from ubiquinone which acts as a lipophilic radical trapping agent within membranes thereby protecting from ferroptosis. Oxidative phosphorylation (OXPHOS) and the tricarboxylic acid (TCA) cycle have both been described to be required for ferroptosis triggered by cystine-depletion or system xc^-^ but not GPX4 inhibition.

**Figure 2 cancers-12-00164-f002:**
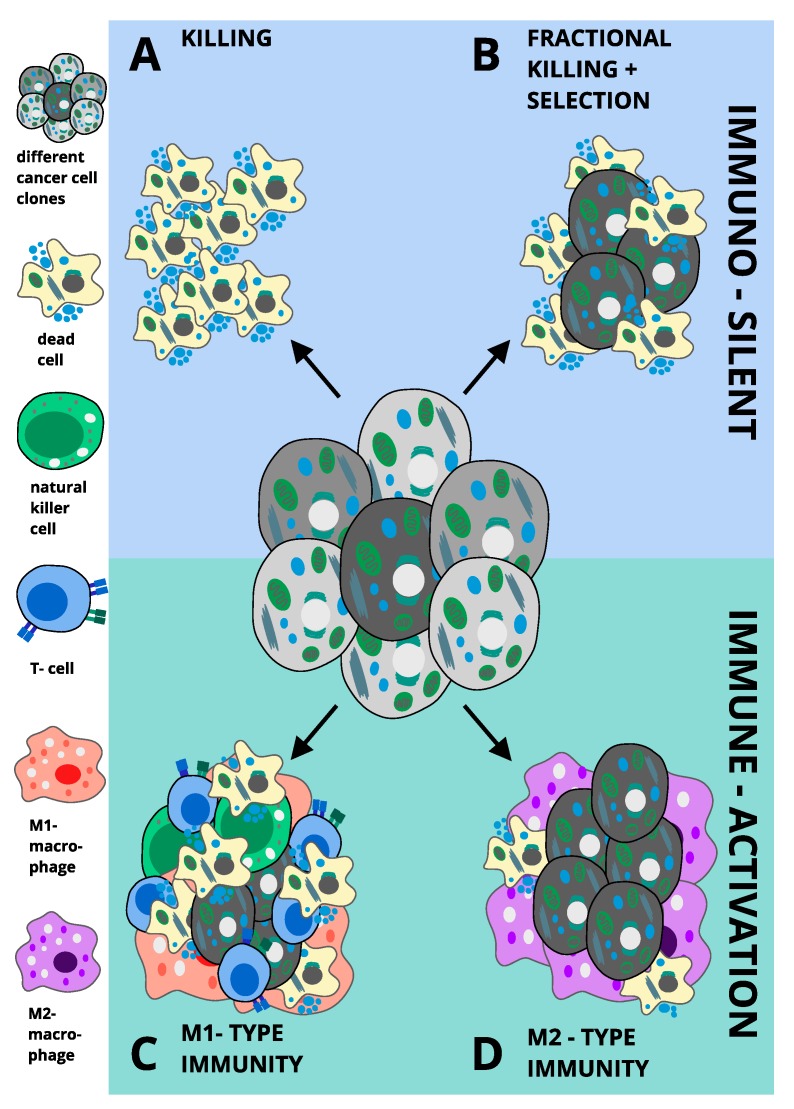
Proposed concepts for the influence of ferroptosis on tumour outcome. (**A**) without raising an immune response, ferroptosis may result in selective and complete killing of tumour cells leading to tumour eradication. (**B**) if immune-silent, ferroptosis may also merely result in fractional killing of cells within a heterogeneous tumour. Over time, this would lead to selection of ferroptosis resistant clones and their outgrowth and overall promotion of tumours. (**C**) if ferroptosis were able to raise an M1-type immune response, M1 macrophages would aid T-cell activation and maintain an anti-tumour immune response resulting in tumour eradication. (**D**) if ferroptosis instead were to raise an M2-type immune response, M2-macrophages would protect tumour cells from T-cell-mediated anti-tumour immune attack, leading to ferroptosis-initiated tumour immune protection and immune escape. Thereby, concept A and C offer a model explaining an anti-tumour effect of ferroptosis, whereas B and D propose model mechanisms for ferroptosis-induced tumour promotion.

**Table 1 cancers-12-00164-t001:** Ferroptosis-inducing drugs.

Reagent	Mechanism of Action	FDA Approved/Clinical Use	Reference
(1S,3R)-RSL3	GPX4 inhibitor	No	[15,34,124,137]
Altretamine	GPX4 inhibitor	Yes/Ovarian cancer treatment	[135,136]
Artesunate	Glutathione S transferase	No/Malaria treatment	[76]
BAY 87-2243	Mitochondrial complex I inhibitor/hypoxia-inducible factor-1 (HIF-1) inhibitor	No	[138,139]
Buthionine sulfoximine (BSO)	γ-GCS inhibitor	No/Clinical trial for neuroblastoma treatment	[34,110,137]
Cyst(e)inase	[Cys] depletion	No	[140]
erastin	System xc- inhibitor	No	
FIN56	Gpx4 degradation/squalene synthase activator	No	[141,142]
Imidazole-ketone erastin	System xc-inhibitor	No	[128]
Piperazine erastin	System xc-inhibitor	No	[15,34]
Sorafenib	System xc-inhibitor	Yes/Renal cell, thyroid, and hepatocellular carcinoma treatment	[38,129,130]
Statins	Block biosynthesis of CoQ10	Yes	[85]
Sulfasalazine	System xc-inhibitor	Yes/Rheumatoid arthritis and inflammatory bowel diseases treatment	[24,131,132,133,134]
Withaferin A	Gpx4 inactivation/Keap1 inactivation	No/Clinical trial for schizophrenia	[75,142]

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
