# Peer review of "Ferroptosis in Cancer Cell Biology"

_cancers, 2020, doi:10.3390/cancers12010164_

Round 1

Reviewer 1 Report

Combine mitochondria induced PUFAs-OOH with LOX to the same side in figure 1 will indicate GPX4 can work on all of these lipid-ROS.

Section 3 (Regulated cell death in cancer) should be moved up before talking about OXPHOS so the paper transitions better.

In Section 2 (OXPHOS as a motor for the development of a cellular antioxidant defense system), there is no introduction for SOD1 and the cellular antioxidant defense system. It would be nice if they included a sentence or so connecting SOD1 to the antioxidant defense system for ROS. 

In Section 4 (Ferroptosis pathway regulation), there should be a short sentence describing what RSL3 is in the last paragraph of Section 4. Also, it would be nice if they mentioned which cancers the cancer cell lines came from as it gives an idea of which cancers can/are generally associated with ferroptosis. 

The last diagram after the conclusion is not very clear. With the Immuno-silent side of the diagram the data lines up clearly. However, on the immune-activation side of diagram seems unclear. I do not know what the different colored objects within the figure mean and what their significance is to the impact of ferroptosis on the tumor. 

Author Response

Point-by-point reply:

We would like to thank all reviewers for taking the time to thoroughly read our manuscript and to provide constructive feedback. We have made every effort to address all points raised and believe that with the resulting changes included in the revised version, our manuscript is now truly improved (see points below).

Reviewer 1

Comments and Suggestions for Authors

Combine mitochondria induced PUFAs-OOH with LOX to the same side in figure 1 will indicate GPX4 can work on all of these lipid-ROS.

We have extensively revised figure 1 and also addressed this constructive point raised (see new figure 1). In addition, we have added a figure legend.

Section 3 (Regulated cell death in cancer) should be moved up before talking about OXPHOS so the paper transitions better.

We thank the reviewer for this very helpful suggestion and have restructured this part accordingly. In the revised manuscript this is now part of the introduction.

In Section 2 (OXPHOS as a motor for the development of a cellular antioxidant defense system), there is no introduction for SOD1 and the cellular antioxidant defense system. It would be nice if they included a sentence or so connecting SOD1 to the antioxidant defense system for ROS. 

We thank the reviewer for pointing this out, this was indeed disconnected. We have now condensed and restructured former chapter 1, 2 and 3 into one introductory chapter covering anti-oxidant defense against ROS (see new chapter 1).

In Section 4 (Ferroptosis pathway regulation), there should be a short sentence describing what RSL3 is in the last paragraph of Section 4. Also, it would be nice if they mentioned which cancers the cancer cell lines came from as it gives an idea of which cancers can/are generally associated with ferroptosis. 

Both edits have been included in the revised text.

The last diagram after the conclusion is not very clear. With the Immuno-silent side of the diagram the data lines up clearly. However, on the immune-activation side of diagram seems unclear. I do not know what the different colored objects within the figure mean and what their significance is to the impact of ferroptosis on the tumor.

We apologize for this mistake and have included a legend describing the cell types represented. In addition, we have included a figure legend describing the 4 alternative proposed scenarios in figure 2.

Reviewer 2 Report

The authors present a well-written and clear review about the roles of ferroptosis in cancer cell biology. I strongly recommend this article for publication and only have minor suggestions:

Authors are suggested to add a short description related to the proposed mechanism/pathway in this diagram under Figures for the readers to easily understand, despite that detail already presented in text. Figure 1 is suggested to add FSP1 and its substrate CoQ10. Figure 2 is suggested to show the meaning for the different color of cells, as well as, the immune cell types in the figure.

In Section 3, the authors are suggested to add more description about the specific characteristics of ferrotposis from other cell death types. Authors are suggested to combine Section 3 into introduction section.

That Nrf2 causes the resistance against ferroptotic induction does not only rely on GSH. It also includes the iron metabolism and intermediate metabolism. Please add them.

There is few studies regarding ferroptosis and immunity, so, the section of ferroptosis and immunity, is most like a discussion of pathology (kidney injure and liver metabolism disorder) and immune. Authors are suggested to add the ferroptosis and immune of cancer cells.

Author Response

Point-by-point reply:

We would like to thank all reviewers for taking the time to thoroughly read our manuscript and to provide constructive feedback. We have made every effort to address all points raised and believe that with the resulting changes included in the revised version, our manuscript is now truly improved (see points below).

Reviewer 2

Comments and Suggestions for Authors

The authors present a well-written and clear review about the roles of ferroptosis in cancer cell biology. I strongly recommend this article for publication and only have minor suggestions:

Authors are suggested to add a short description related to the proposed mechanism/pathway in this diagram under Figures for the readers to easily understand, despite that detail already presented in text. Figure 1 is suggested to add FSP1 and its substrate CoQ10. Figure 2 is suggested to show the meaning for the different color of cells, as well as, the immune cell types in the figure.

We appreciate the constructive feedback. We have extensively revised figure 1 and 2, have added figure legends for both and the missing pathway components.

In Section 3, the authors are suggested to add more description about the specific characteristics of ferrotposis from other cell death types. Authors are suggested to combine Section 3 into introduction section.

We have taken this reviewer’s very helpful advice and have combined section 1, 2 and 3 into a single introductory paragraph. As we have had space constrains, we have here referred to a review covering the nature of other types of regulated cell death for further reading on their respective mechanisms.

That Nrf2 causes the resistance against ferroptotic induction does not only rely on GSH. It also includes the iron metabolism and intermediate metabolism. Please add them.

This point is very well taken and we have now included this point.

There is few studies regarding ferroptosis and immunity, so, the section of ferroptosis and immunity, is most like a discussion of pathology (kidney injure and liver metabolism disorder) and immune. Authors are suggested to add the ferroptosis and immune of cancer cells. 

We have changed the title of this chapter to fit this reviewer’s suggestion. By also changing the subtitle for the subchapter and including a some clarifying sentences and an additional source we hope to have proposed potential effects of ferroptosis within the tumor microenvironment.

Reviewer 3 Report

This is an excellent and very timely review on a topic that has raised a high importance in the last years. The manuscript summaries the studies about ferroptosis and cancer in a very concise way. I suggest only two addresses:

Line 324: in what organs was GPX4 depleted?

Liner 330: Could the authors speculate about the relation between GPX4 and necroptosis activation

A table summarizing the pre-clinical studies with ferroptosis inductors for cancer treatment would help us to have a global vision of its clinical potential.

Author Response

Point-by-point reply:

We would like to thank all reviewers for taking the time to thoroughly read our manuscript and to provide constructive feedback. We have made every effort to address all points raised and believe that with the resulting changes included in the revised version, our manuscript is now truly improved (see points below).

Reviewer 3

This is an excellent and very timely review on a topic that has raised a high importance in the last years. The manuscript summaries the studies about ferroptosis and cancer in a very concise way. I suggest only two addresses:

Line 324: in what organs was GPX4 depleted?

We thank the reviewer for pointing this out. The missing information has now been added to the revised version of the manuscript (inducible whole-body deletion).

Liner 330: Could the authors speculate about the relation between GPX4 and necroptosis activation

We have added the following additional information into the revised manuscript:

“Mechanistically, the authors show that GPX4 deletion leads to glutathionylation and inactivation of caspase 8, which triggers necroptosis independently of TNF”.  

A table summarizing the pre-clinical studies with ferroptosis inductors for cancer treatment would help us to have a global vision of its clinical potential.

This is an excellent suggestion. Such a table has now been included as table 1 in the revised version of the manuscript.

Reviewer 4 Report

This review focuses on basic mechanisms of ferroptosis signaling and its involvement in tumor suppression and cancer immunology in the context cancer therapy.

General: Although the review provide detailed timely information on the topic, the review is difficult to read, and the rationale of its structure is not always evident. Furthermore, the relevance of the Figures is unclear.

Specific:

Introduction: The “introduction” is not an introduction to the topic of the review, namely cancer biology and ferroptosis as a distinct mode of cell death. Rather, the heading of the 2nd paragraph appears more appropriate to summarize that text. Introduction: In their introduction, the authors should emphasize to which extent this review represents a novelty or an extension of recently published reviews on the topic. At least 5 major reviews on the topic have been published in 2019 (1. Redox biology of regulated cell death in cancer: A focus on necroptosis and ferroptosis. Florean C, Song S, Dicato M, Diederich M. Free Radic Biol Med. 2019 Apr;134:177-189; 2. Ferroptosis is a type of autophagy-dependent cell death. Zhou B, Liu J, Kang R, Klionsky DJ, Kroemer G, Tang D. Semin Cancer Biol. 2019 Mar 14. pii: S1044-579X(19)30006-9; 3. Ferroptosis at the crossroads of cancer-acquired drug resistance and immune evasion. Friedmann Angeli JP, Krysko DV, Conrad M. Nat Rev Cancer. 2019 Jul;19(7):405-414; 4. Targeting Ferroptosis to Iron Out Cancer. Hassannia B, Vandenabeele P, Vanden Berghe T. Cancer Cell. 2019 Jun 10;35(6):830-849; 5. Recent Progress in Ferroptosis Inducers for Cancer Therapy. Liang C, Zhang X, Yang M, Dong X. Adv Mater. 2019 Dec;31(51):e1904197). And I am sure that more groundbreaking reviews on the topic have been published as well in previous years. It is good scientific practice to acknowledge previous reviews covering the topic. Otherwise, the authors suggest that their review is unique in the field, which is not the case. Introduction: The authors neglect that many cancer cells do not perform OXPHOS, but rather ferment (Warburg effect). This should be discussed. The 2nd paragraph has nothing to do with OXPHOS. Rather, it describes the cellular redox processes and the contribution of iron. The heading should be changed accordingly. Lines 167-171: It is not clear from the text or from Figure 1 which important functions GPX4 exerts downstream of mitochondria whereas upstream functions of GPX4 are more understandable from the text. Figure 1 does not really reflect the interactions between the Fenton reaction, ferroptosis signaling and mitochondria. Rather, the Figure seems to describe three parallel pathways that converge onto (cell ?) membrane lipids . Is it what the authors intend to describe in that Figure and does it reflect the text? Line 368: Omit parentheses from NRF2. Figure 2 is not helpful at all. In this “artwork”, ferroptosis and tumor outcome are not described at all. Where is ferroptosis described? What do the different colors and shapes of cells represent?  

Author Response

Point-by-point reply:

We would like to thank all reviewers for taking the time to thoroughly read our manuscript and to provide constructive feedback. We have made every effort to address all points raised and believe that with the resulting changes included in the revised version, our manuscript is now truly improved (see points below).

Reviewer 4

This review focuses on basic mechanisms of ferroptosis signaling and its involvement in tumor suppression and cancer immunology in the context cancer therapy.

General: Although the review provide detailed timely information on the topic, the review is difficult to read, and the rationale of its structure is not always evident. Furthermore, the relevance of the Figures is unclear.

Specific:

Introduction: The “introduction” is not an introduction to the topic of the review, namely cancer biology and ferroptosis as a distinct mode of cell death. Rather, the heading of the 2nd paragraph appears more appropriate to summarize that text. Introduction: In their introduction, the authors should emphasize to which extent this review represents a novelty or an extension of recently published reviews on the topic. At least 5 major reviews on the topic have been published in 2019 (1. Redox biology of regulated cell death in cancer: A focus on necroptosis and ferroptosis. Florean C, Song S, Dicato M, Diederich M. Free Radic Biol Med. 2019 Apr;134:177-189; 2. Ferroptosis is a type of autophagy-dependent cell death. Zhou B, Liu J, Kang R, Klionsky DJ, Kroemer G, Tang D. Semin Cancer Biol. 2019 Mar 14. pii: S1044-579X(19)30006-9; 3. Ferroptosis at the crossroads of cancer-acquired drug resistance and immune evasion. Friedmann Angeli JP, Krysko DV, Conrad M. Nat Rev Cancer. 2019 Jul;19(7):405-414; 4. Targeting Ferroptosis to Iron Out Cancer. Hassannia B, Vandenabeele P, Vanden Berghe T. Cancer Cell. 2019 Jun 10;35(6):830-849; 5. Recent Progress in Ferroptosis Inducers for Cancer Therapy. Liang C, Zhang X, Yang M, Dong X. Adv Mater. 2019 Dec;31(51):e1904197). And I am sure that more groundbreaking reviews on the topic have been published as well in previous years. It is good scientific practice to acknowledge previous reviews covering the topic. Otherwise, the authors suggest that their review is unique in the field, which is not the case.

We appreciate the constructive feedback and have re-structured the introduction according to this reviewer’s and the other reviewer’s suggestion for a clearer narrative. As per suggestion, we have now also added in the above-named reviews. It was by no means our intention to give the impression suggested but rather to provide a timely update integrating the most recent findings on ferroptosis and cancer with our stand-point.

Introduction: The authors neglect that many cancer cells do not perform OXPHOS, but rather ferment (Warburg effect). This should be discussed.

We thank the reviewer for this important point. This discussion has now been added to the introduction chapter.

The 2nd paragraph has nothing to do with OXPHOS. Rather, it describes the cellular redox processes and the contribution of iron. The heading should be changed accordingly.

In the revised version of the manuscript, we have merged the former chapter 1-3 and re-structured it. We think that the suggested re-structuring has truly improved the text-flow.

Lines 167-171: It is not clear from the text or from Figure 1 which important functions GPX4 exerts downstream of mitochondria whereas upstream functions of GPX4 are more understandable from the text.

We apologize for this imprecision, we have now clarified these points in the text and in new figure 1 and figure legend 1.

Figure 1 does not really reflect the interactions between the Fenton reaction, ferroptosis signaling and mitochondria. Rather, the Figure seems to describe three parallel pathways that converge onto (cell ?) membrane lipids . Is it what the authors intend to describe in that Figure and does it reflect the text?

We acknowledge that this was not clear before. We have therefore thoroughly revised figure 1 to clarify these points. Moreover, we have added a figure legend.

Line 368: Omit parentheses from NRF2.

Has been removed.

Figure 2 is not helpful at all. In this “artwork”, ferroptosis and tumor outcome are not described at all. Where is ferroptosis described? What do the different colors and shapes of cells represent?  

We agree that figure 2 was lacking a legend describing the cell types and was therefore not clear. We have now added these to the image and have added a structured figure legend describing the 4 proposed tumour outcomes.

Round 2

Reviewer 4 Report

All my comments have been addressed in a satisfactory manner.

Please, rotate Figure 2 clockwise by 90o.

Author Response

Figure 2 has been revised accordingly.